# The First Homozygote Mutation c.499G>T (Asp167Tyr) in the *RPE65* Gene Encoding Retinoid *Isomerohydrolase* Causing Retinal Dystrophy

Mirjana Bjeloš [1,2,3], Ana Ćurić [1,3], Benedict Rak [1], Mladen Bušić [1,2,3,*] and Biljana Kuzmanović Elabjer [1,2,3]

1    University Eye Department, Reference Center of the Ministry of Health of the Republic of Croatia for Inherited Retinal Dystrophies, Reference Center of the Ministry of Health of the Republic of Croatia for Pediatric Ophthalmology and Strabismus, University Hospital "Sveti Duh", 10000 Zagreb, Croatia
2    Faculty of Medicine, Josip Juraj Strossmayer University of Osijek, 31000 Osijek, Croatia
3    Faculty of Dental Medicine and Health Osijek, Josip Juraj Strossmayer University of Osijek, 31000 Osijek, Croatia
*    Correspondence: mbusic@kbsd.hr

**Abstract:** RPE65, an abundant membrane-associated protein present in the retinal pigment epithelium (RPE), is a vital retinoid isomerase necessary for regenerating 11-*cis*-retinaldehyde from *all-trans* retinol in the visual cycle. In patients with inherited retinal dystrophy (IRD), precise genetic diagnosis is an indispensable approach as it is required to establish eligibility for the genetic treatment of *RPE65*-associated IRDs. This case report aims to report the specific phenotype–genotype correlation of the first patient with a homozygous missense variant *RPE65* c.499G>T, p. (Asp167Tyr). We report a case of a 66-year-old male who demonstrated a unique phenotype manifesting less severe functional vision deterioration in childhood and adolescence, and extensive nummular pigment clusters. The underlying causes of the differences in the typical bone spicule and atypical nummular pigment clumping are unknown, but suggest that the variant itself influenced the rate of photoreceptor death. Functional studies are needed to define whether the substitution of aspartate impairs the folding of the tertiary RPE65 structure only and does not lead to the complete abolishment of chromophore production, thus explaining the less severe phenotype in adolescence.

**Keywords:** retinal dystrophies; retinal pigments; melanins; genetic therapy

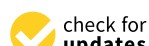



## 1. Introduction

*RPE65*, an abundant membrane-associated protein present in the retinal pigment epithelium (RPE), is a vital retinoid isomerase necessary for regenerating 11-*cis*-retinaldehyde from *all-trans* retinol in the visual cycle [1]. *RPE65* variants account for 2.1% of patients with autosomal recessive inherited retinal dystrophies (IRDs), and the phenotype resulting from *RPE65* variants appears to be relatively uniform [2]. More than 240 variants of *RPE65* are currently annotated as disease-causing in the HGMD Professional Variant Database (version 2022.2) [3]. Half of the variants are clearly missense variants, and approximately 40% are truncating variants (nonsense, frameshift, variants affecting splicing, and gross deletions). The loss of *RPE65* function involves the loss of catalytic activity, a lower expression level of *RPE65*, or the rapid degradation of the variant protein [4,5].

In IRD patients, precise genetic diagnosis is an indispensable approach, as it is required to establish eligibility for the genetic treatment of *RPE65*-associated IRDs with voretigene neparvovec: a recombinant adeno-associated viral vector providing a functional *RPE65* gene to act in place of a mutated *RPE65* gene [6].

This case report aims to report the specific phenotype–genotype correlation of the first patient with a homozygous missense variant *RPE65* c.499G>T, p. (Asp167Tyr), namely nummular pigment deposits. We hypothesized the *fuel* theory to reveal the mechanisms

behind the development of typical bone spicule and atypical nummular pigment clumping. We assume that the RPE detachment and migration to perivascular inner retinal sites is caused by the need for *fuel* intake, as RPE cells cannot regenerate 11-*cis* retinal from *all-trans* retinol.

## 2. Case Presentation

A 66-year-old male with IRD was referred to our Eye Department for clinical examination and genetic testing. The patient manifested nyctalopia since early childhood. However, he attended elementary school, high school, and college according to the regular program. At the age of 17, a more pronounced decline in visual functions ensued. His brother was also visually impaired, but his family history was otherwise unremarkable.

On clinical examination, his best corrected visual acuity (BCVA) was light perception on the right eye (RE) and amaurosis on the left eye (LE). The CSV-1000 contrast sensitivity test, standardized color vision tests (Farnsworth's D-15 dichotomous test and Lanthony desaturated 15-hue panel), Octopus static and kinetic perimetry (Haag-Streit Inc., Mason, OH, USA), and MAIA microperimetry (iCare Finland Oy, Vantaa, Finland) were not feasible due to the low BCVA.

HRA+ OCT Spectralis® (Heidelberg Engineering, Heidelberg, Germany) depicted prominent sub-RPE deposits and epiretinal membranes on both eyes (BE) with vitreoretinal traction on LE (Figure 1). Central macular thicknesses of 253 and 323 microns were measured on the RE and LE, respectively. An ONL of 32 microns was present on the RE.

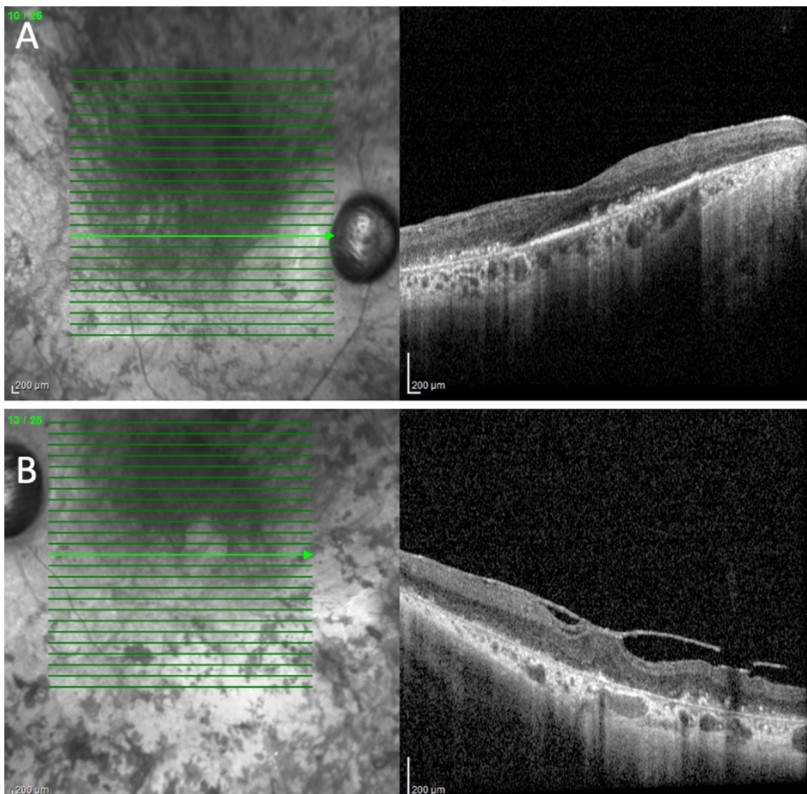

**Figure 1.** Spectral domain OCT scan (HRA+ OCT Spectralis®) of the right (**A**) and left (**B**) eyes depicting prominent sub-RPE deposits and epiretinal membranes on both eyes with vitreoretinal traction on the left eye.

The Optos® California (Optos Inc., Marlborough, MA, USA) ultra-widefield imaging depicted a pale and waxy optic nerve head with clear boundaries, more pronounced on his LE. In the posterior pole, extensive chorioretinal atrophy was noted, while in the mid- and far peripheries, clumped pigmentary changes in the nummular appearance were present.

Retinal vessels presented obliterative sclerosis, revealing avascular mid- and far peripheries. The Optos® short-wavelength fundus autofluorescence was absent (Figure 2).

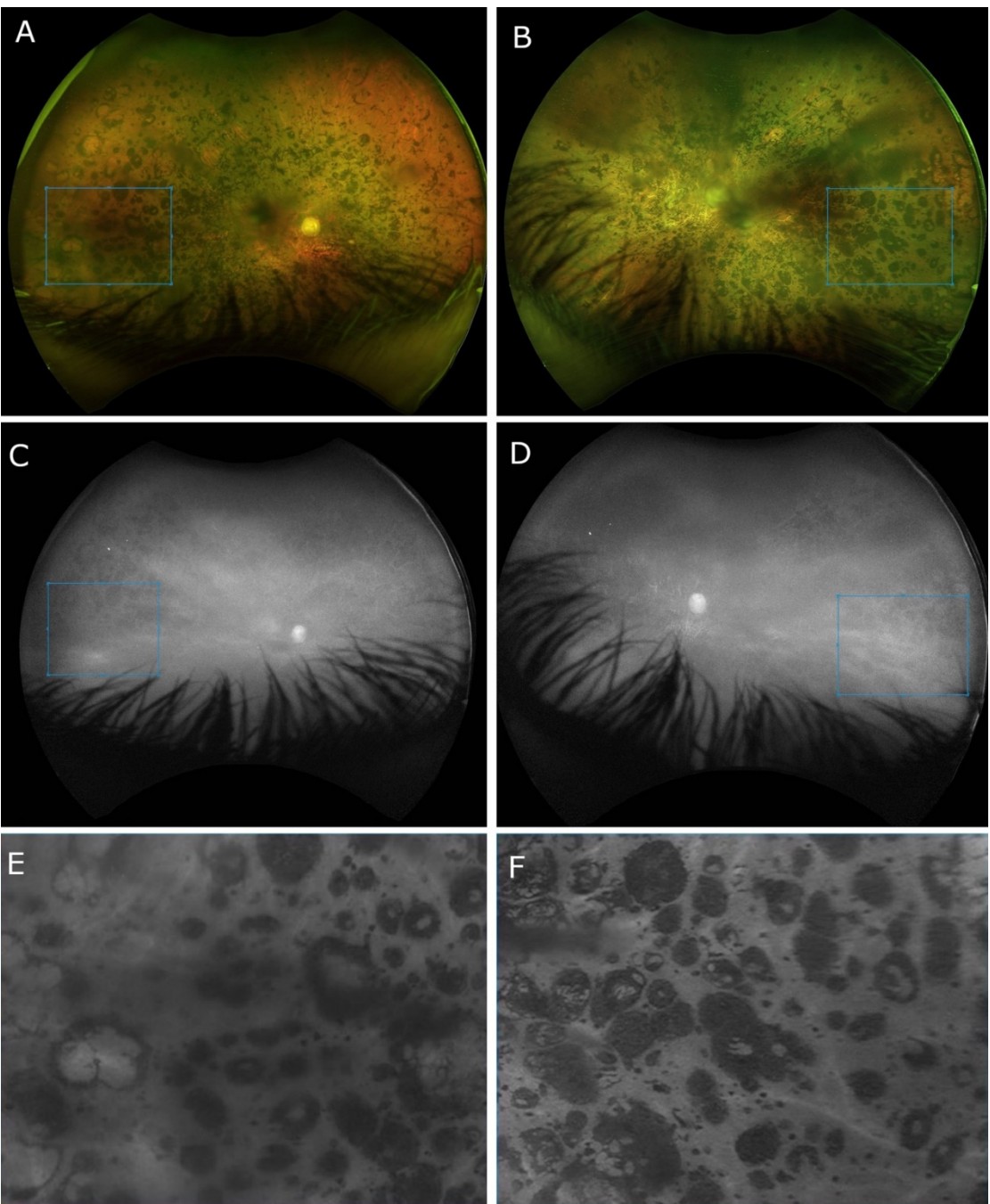

**Figure 2.** Ultra-widefield (UWF) 200° fundus photos of the right eye (**A**) and left eye (**B**) showing nummular pigment clumping of variable sizes. Retinal vessels are presented as solid strands of hyaline-like connective tissue with avascular mid- and peripheral retina. Short-wavelength fundus autofluorescence (Optos® California) UWF 200° photos of the right eye (**C**) and left eye (**D**) depicting the absence of autofluorescence. Chorioretinal atrophy revealed the hyperautofluorescent attributes of the sclera. Enlarged image of the areas (marked with the blue squares in (**A**–**D**)) with nummular pigment clumping on RE (**E**) and LE (**F**) obtained with a 100% red blend and 4.35× magnification.

The full-field electroretinography testing (Roland Consult, RETI-port/scan 21, Roland Consult Stasche and Finger GmbH–German Engineering, Brandenburg an der Havel,

Germany) according to ISCEV standards exhibited extinguished scotopic and photopic responses on BE.

IOLMaster® 700 (Zeiss, Oberkochen, Germany) optical biometry measured axial length of 22.83 mm RE and 22.66 mm LE, while retinoscopy was unremarkable (+0.75 Dsph on RE, and +0.75 Dsph/−1.00 Dcyl ax 115° on LE).

Full-field stimulus testing (Metrovision, Perenchies, France) detected responses of 43 dB, 26 dB, and 53 dB to white, red, and blue light, respectively, on the RE, while on the LE, no response was identified.

Given the clinical diagnosis of IRD, the patient was referred to genetic testing. Sequence analysis using the Blueprint Genetics Retinal Dystrophy Panel identified a homozygous missense variant *RPE65* c.499G>T, p. (Asp167Tyr).

### 3. Discussion

*RPE65* is the key enzyme of the classical visual cycle [7], a complex enzymatic pathway of *trans*-to-*cis* isomerization, regenerating the visual chromophore [8].

The basic *RPE65* structure is a seven-bladed β-propeller with single-strand extensions on blades VI and VII and a two-strand extension on blade III. It seems that a majority of the amino acid substitutions found in LCA and RP patients occur within or adjacent to the blades of the β-propeller, rather than in the connecting loops and helices.

Position 167 resides within strand 11 of blade I, away from the iron ion active site cavity [9]. The iron ion binding site is coupled with four His and three Glu residues, each derived from a single blade. As a hydrophilic residue, Asp167 could serve as a gatekeeper protecting the aggregation-prone regions of proteins [10,11]. Thus, Asp167Tyr replacement could allow the formation of predominantly hydrophobic contacts, as observed with D477G and induced toxic gain-of-function [9].

To the best of our knowledge, this is the first patient reported in the literature to be homozygous for *RPE65* c.499G>T, p. (Asp167Tyr) variant. In the Genome Aggregation Database (gnomAD), the p.Asp167Tyr variant was absent from 50 control individuals, but was reported at a frequency of 0.000026 in the European (non-Finnish) population [12]. The variant affects a highly conserved amino acid in the carotenoid oxygenase domain of the protein; there was a large physicochemical difference between asparagine and tyrosine (Grantham score 160, [0–215]), and all of the in silico tools utilized (PolyPhen, SIFT) predicted the alteration to be deleterious [12].

This variant is a missense variant that has been reported in a compound heterozygous state in three individuals with either LCA or RP [2,13–15]. Further, the p.Asp167Tyr variant was also identified in a heterozygous state in one additional case in whom a second variant was not identified [16]. Based on the collective evidence, the *RPE65* c.499G>T, p. (Asp167Tyr) variant has been classified as pathogenic/likely pathogenic for *RPE65*-related disorders [17].

Many *RPE65* patients have similar phenotypes characterized by poor, but useful, visual function in early life that declines significantly throughout school age. Additionally, some of these patients retain residual, although significantly narrowed, islands of peripheral vision into the third decade of life [2]. As presented by Chung et al., 88.5% of affected individuals have severely reduced or absent fundus autofluorescence [18]. The majority of *RPE65* patients have whitish deposits and scarce pigmentary changes in the early stages and typical RP bone spicule-like pigment deposits later in the disease course, with a spectrum of disease severity that ranges from a mild form to a severe one [19–22]. Almost all of the mutations in *RPE65* are inherited in a recessive manner; however, an autosomal dominant form of *RPE65*-associated RP with choroidal involvement has also been identified. Carriers of the D477G mutation, unlike typical RP patients, exhibit areas of retinal atrophy ranging from the choroid to the photoreceptor layer, and initially present with a central visual defect. Further, choroidal atrophy and RPE hypertrophy are frequently observed and distinctive features associated with this mutation [8].

Our patient manifested a unique genotype and phenotype biallelic *RPE65* c.499G>T, p. (Asp167Tyr) variant that presented with less severe functional vision deterioration in

childhood and extensive nummular pigment clusters affecting the mid- and far peripheries, reported only in patients with chloridemia, retinitis pigmentosa 86 (RP-86), and retinitis pigmentosa 87 (RP-87) with choroidal involvement [23–26].

The underlying causes of the differences in the typical bone spicule and atypical nummular pigment clumping are unknown, but suggest that the variant itself influences the rate of photoreceptor death.

The major stimulus that triggers RPE detachment from Bruch's membrane and the migration of RPE cells to perivascular sites in the inner retina is proposed to be the direct contact of deep capillary retinal vessels and apical side of the RPE due to photoreceptor loss [27]. Bone spicule pigment clusters delineate retinal capillaries and venules at different retinal layers from the RPE to the retinal nerve fiber layer [28]. Relocated RPE cells induce endothelial fenestration, deposit the extracellular matrix, and seal the vessel by forming tight junctions reestablishing the blood–retinal barrier comparable to the choriocapillaris/RPE interface [27,29]. It is hypothesized that RPE cells are attracted to vascular basal lamina due to the endothelial release of cytokines [28,29] and higher intravascular oxygen levels [28], causing a vicious cycle of secondary retinal obliterative sclerosis. However, we assume that the RPE detachment and migration to perivascular inner retinal sites is caused by the need for *fuel* intake, as RPE cells cannot regenerate 11-*cis* retinal from *all-trans* retinol; thus, they need to fuel themselves from outer sources i.e., retinal capillaries. In a healthy retina, *all-trans* retinol is taken up from the bloodstream of the fenestrated choriocapillaris to the RPE. Thus, to provide supply of the chromophore, the RPE detaches, migrates, and induces the pathologic fenestration of the non-fenestrated retinal capillaries.

In contrast to typical bone spicule clusters of relocated RPE cells containing few melanin granules in the inner retina, the rarer form of rounded nummular flecks demonstrates a different pattern of reactive changes with densely packed melanin granules clustered against Bruch's membrane [30]. We hypothesize the latter scenario to occur in cases of a lower expression level of *RPE65* or the rapid degradation of the variant protein, as observed in D477G *RPE65* mutation [8]. The study of missense variant *RPE65* c.1430A>G, p. (Asp477Gly) was reported to cause a dominantly inherited, distinctive form of RP, RP-87 with choroidal involvement, evidencing the abnormally slow regeneration of the 11-*cis* retinal chromophore [31]. This corroborates the hypothesized *fuel theory* because, in RP-87 induced by D477G *RPE65* substitution, toxic gain-of function is induced, the chromophore is slowly renewed, enabling RPE cells to stay in situ, thus presenting with a less severe phenotype in adolescence [31]. Heavily pigmented, they form nummular deposits against Bruch's membrane, possibly consistent with the hexagonal lobules of choriocapillaris. Although RPE is generally viewed as a postmitotic cell, the concept that melanin turnover is absent in the adult RPE is grounded in the tyrosinase activity identified only in prenatal periods [32] and is absent much before gestation ends [33]. Whether the RPE cells in RP are capable of melanin synthesis is still unknown. It has been evidenced that tyrosinase can be induced in postnatal RPE by the orthodenticle homeobox 2 (OTX2) activation of the human tyrosinase gene promoter [34], overexpression of tyrosinase [35] and the transcriptional repressor Zeb1 [36], and phagocytosis of rod outer segments [37], demonstrating increased pigmentation.

With aging, the RPE melanin, a powerful electron acceptor, may undergo significant chemical modifications induced by oxygen, light, and toxic ions that bind to melanin, converting its antioxidant potential to phototoxic, pro-oxidative potential [38,39]. It seems plausible that this prooxidative potentially advances chorioretinal atrophy later in the disease's course.

Our patient is homozygous for the *RPE65* c.499G>T, p. (Asp167Tyr) variant, consistent with autosomal recessive inheritance. Functional studies are needed to define whether the substitution of Asp impairs the folding of the tertiary *RPE65* structure only [40] and does not lead to the complete abolishment of chromophore production.

## 4. Conclusions

In conclusion, *RPE65* c.499G>T, p. (Asp167Tyr) is classified as pathogenic, based on currently available evidence supporting its disease-causing role. Diseases caused by *RPE65* c.499G>T, p. (Asp167Tyr) are inherited in an autosomal recessive manner. To the best of our knowledge, this is the first patient reported as homozygous for this variant. The patient demonstrated a unique phenotype not previously described in the literature, confirming that *RPE65*-associated IRDs may pose special challenges for genotype/phenotype correlations [18].

**Author Contributions:** Conceptualization: M.B. (Mirjana Bjeloš), A.Ć. and B.R.; methodology: M.B. (Mirjana Bjeloš), A.Ć., B.R., M.B. (Mladen Bušić) and B.K.E.; validation: M.B. (Mirjana Bjeloš), A.Ć., B.R., M.B. (Mladen Bušić) and B.K.E.; formal analysis: M.B.(Mirjana Bjeloš), A.Ć., B.R. and M.B. (Mladen Bušić); investigation, resources, and data curation: M.B. (Mirjana Bjeloš), A.Ć., B.R., M.B. (Mladen Bušić) and B.K.E.; writing—original draft preparation: M.B. (Mirjana Bjeloš), A.Ć. and M.B. (Mladen Bušić); writing—review and editing: M.B. (Mirjana Bjeloš), A.Ć., B.R., M.B. (Mladen Bušić) and B.K.E.; supervision: M.B. (Mirjana Bjeloš), A.Ć., M.B. (Mladen Bušić) and B.K.E. All authors have read and agreed to the published version of the manuscript.

**Funding:** No funding was received for this research.

**Institutional Review Board Statement:** Not applicable.

**Informed Consent Statement:** Prior to testing, written informed consent was obtained from the patient. No compensation or incentive was offered to the subject to participate in the study.

**Data Availability Statement:** The data presented in this case report are available on request from the corresponding author. The data are not publicity available due to privacy protection.

**Conflicts of Interest:** The authors declare no conflict of interest.

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
