# Peer review of "The First Homozygote Mutation c.499G>T (Asp167Tyr) in the RPE65 Gene Encoding Retinoid Isomerohydrolase Causing Retinal Dystrophy"

_cimb, doi:10.3390/cimb44120436_

Round 1

Reviewer 1 Report

I have no recommendations for the author collective.

Reviewer 2 Report

Dear Mirjana Bjelos and colleagues,

I did find the case report on the homozygous missense mutation in the RPE65 gene encoding Retinoid Isomerohydrolase interesting.

Here are my comments:

The title is not very helpful as the case report doesn´t say anything about the “fuel theory”. It only reports the clinical characteristics of a patient. My suggestion would be something along the lines of “The first homozygote mutation c.499G>T (Asp167Tyr) in the RPE65 gene encoding Retinoid Isomerohydrolase causing retinal dystrophy.”

Overall, the text is difficult to read as it is clearly written only for a small expert group. It also is in need of some English editing.

Abstract:

The function of the RPE65 protein and its role in the visual cycle should be clearly stated at the beginning of the abstract. The reference to the fuel theory should be deleted. This can become part of the discussion. In line 24 it should say aspartate not asparagine

Introduction & Case:

L33: please use full name of IRD here

L: 46: please explain the reader here what the fuel theory is about that will be discussed later

L:58-60 this sentence is unclear

I would like to suggest including a display item showing the location of Asp167Tyr replacement in the protein structure of the RPE65 enzyme in relation to its active site. Please see PDB ID code 3FSN (https://www.pnas.org/doi/pdf/10.1073/pnas.0906600106)

Discussion:

Currently the discussion is a second introduction meaning that it summarises relevant information but without a direct link to the possible implications of the Asp167Tyr mutation (hence the structure image).

It may help to remind the reader about the role of the enzyme in the visual cycle and the consequences of mutation in it (i.e. move the relevant sections from the second half of the discussion to its start).

Is the Asp167Tyr mutation already known in a heterozygote patient? Please comment on this.

How dos the phenotypes of this case compare with other mutations like D447G (Insights into the pathogenesis of dominant retinitis pigmentosa associated with a D477G mutation in RPE65 - PubMed (nih.gov))? Please comment on the phenotypes of other Mutations in the RPE65 enzyme in the discussion.

Please discuss the possible cause for the pigment aggregates in the presence of the mutated enzyme more clearly to establish a stronger linkage to the case.

L116 …different combination of RPE65 mutations in one heterozygote patient? This is not clear.

L129 please use full name retinitis pigmentosa-86 (and please explain the reader what the numbers refer to)

Round 2

Reviewer 2 Report

Dear Mirjana Bjelos and colleagues,

thank you very much for the thorough responses to my comments. The manuscript is now much improved and accessible for a wider readership.